# New Challenging Systemic Therapies for Juvenile Scleroderma: A Comprehensive Review

**DOI:** 10.3390/ph18050643

**Published:** 2025-04-28

**Authors:** Chiara Sassetti, Claudia Borrelli, Martha Mazuy, Cristina Guerriero, Donato Rigante, Susanna Esposito

**Affiliations:** 1Pediatric Clinic, Department of Medicine and Surgery, University of Parma, 43126 Parma, Italy; chiara.sassetti@unipr.it (C.S.); claudia.borrelli@unipr.it (C.B.); martha.mazuy@unipr.it (M.M.); 2Unit of Dermatology, Fondazione Policlinico Universitario A. Gemelli IRCCS, 00168 Rome, Italy; cristina.guerriero@unicatt.it; 3Department of Life Sciences and Public Health, Fondazione Policlinico Universitario A. Gemelli IRCCS, 00168 Rome, Italy; donato.rigante@unicatt.it; 4Università Cattolica Sacro Cuore, 00168 Rome, Italy

**Keywords:** juvenile scleroderma, systemic sclerosis, childhood, personalized medicine, innovative biotechnologies

## Abstract

**Background:** Juvenile scleroderma (JS) comprises a group of rare chronic autoimmune and fibrosing disorders in children, primarily presenting as juvenile localized scleroderma (jLS) or juvenile systemic sclerosis (jSS). While jLS predominantly affects the skin and subcutaneous tissues, jSS may involve multiple internal organs and is associated with increased morbidity and mortality. Due to the scarcity of pediatric-specific clinical trials, the current treatment strategies are largely empirical and often adapted from adult protocols. **Objective:** This narrative review aims to provide a comprehensive update on emerging systemic therapies for juvenile scleroderma, focusing on biologics, small molecule inhibitors, and advanced cellular interventions, to support the development of more personalized and effective pediatric treatment approaches. **Methods:** A literature search was conducted through PubMed and a manual bibliographic review, covering publications from 2001 to 2024. Only English-language studies involving pediatric populations were included, comprising randomized controlled trials, reviews, and case reports. Additional searches were performed for drugs that are specifically used in juvenile scleroderma. **Results**: Biologic agents such as tocilizumab, rituximab, and abatacept, along with small molecules including Janus kinase (JAK) inhibitors and imatinib, have demonstrated potential in managing refractory cases by reducing skin fibrosis and pulmonary involvement. Novel approaches—such as pamrevlumab, nintedanib, and chimeric antigen receptor (CAR-T) cell therapy—target fibrotic and autoimmune pathways but remain investigational in children. Autologous stem cell transplantation (ASCT) has also been explored in severe, treatment-resistant cases, although data are extremely limited. The overall evidence base is constrained by small sample sizes, a lack of controlled pediatric trials, and reliance on adult extrapolation. **Conclusions:** While innovative systemic therapies show promise for juvenile scleroderma, their widespread clinical application remains limited by insufficient pediatric-specific evidence. Large, multicenter, long-term trials are urgently needed to establish safety, efficacy, and optimal treatment algorithms that are tailored to the pediatric population.

## 1. Introduction

Juvenile scleroderma (JS) encompasses a group of rare, chronic fibrosing disorders that arise during childhood, primarily characterized by progressive collagen deposition leading to skin hardening and eventual tissue fibrosis. The condition manifests in two main clinical forms, juvenile localized scleroderma (jLS) and juvenile systemic sclerosis (jSS) [1], with jLS occurring approximately 6 to 10 times more frequently than jSS [2]. Although both conditions share overlapping pathogenic pathways, they are now recognized as distinct entities rather than parts of a disease continuum, as jLS does not evolve into jSS [3].

The clinical burden of JS can be profound. While mortality is rare in jLS, children may experience serious complications such as disfigurement, joint contractures, and reduced mobility, significantly affecting their quality of life. jSS, on the other hand, carries a higher risk of life-threatening organ involvement, particularly of the lungs, heart, and gastrointestinal system, resulting in substantially greater long-term morbidity and mortality [3].

jLS presents in several subtypes depending on the extent and depth of skin involvement, including (a) circumscribed morphea, (b) generalized morphea, (c) linear scleroderma, (d) pansclerotic morphea, and (e) mixed forms combining features of multiple subtypes [4]. Linear scleroderma, which is common in pediatric cases, may extend beyond the dermis into muscles and bones, leading to skeletal abnormalities. The most severe variant, pansclerotic morphea, involves full-thickness skin changes across large body areas and carries the risk of chronic ulcers and cutaneous malignancies, although internal organs are typically spared.

In contrast, jSS typically begins with swelling of the fingers, progressing to Raynaud’s phenomenon and widespread fibrosis. Organ involvement is common: pulmonary disease, including interstitial lung disease and pulmonary hypertension, affects up to half of patients and is a leading cause of death. Muscle atrophy, gastrointestinal dysfunction (often presenting as gastroesophageal reflux), and, less frequently, cardiac or renal complications may also occur. Central nervous system involvement appears more prevalent in pediatric jSS than in adult forms [5].

To standardize diagnosis and facilitate research, a multicenter collaboration involving the Pediatric Rheumatology European Society (PRES), American College of Rheumatology (ACR), and European Alliance of Associations for Rheumatology (EULAR) developed classification criteria for jSS in 2007. These criteria require one major criterion and at least two of twenty minor criteria, spanning various clinical domains and laboratory findings (Table 1) [6].

Although the exact pathogenesis of juvenile scleroderma remains incompletely understood, both jLS and jSS involve complex interactions between genetic predisposition, environmental triggers, autoimmunity, and immune dysregulation [7]. Shared features include the activation of both innate and adaptive immune responses, leading to fibroblast stimulation and collagen overproduction. Key inflammatory and profibrotic mediators include transforming growth factor (TGF)-β, platelet-derived growth factor (PDGF), connective tissue growth factor (CTGF), interleukins (IL-6, IL-4, IL-1α), and chemokine CXCL4, all of which contribute to chronic inflammation, fibrosis, and vascular dysfunction [8].

However, important pathophysiological differences also exist. jSS is more frequently associated with specific HLA alleles and a broader range of circulating autoantibodies, reflecting systemic immune activation and a more robust autoimmune profile compared to the more localized, dermal-limited inflammation seen in jLS. These differences underscore the need for divergent therapeutic approaches targeting distinct immune pathways. Moreover, in jLS, accelerated immune maturation and heightened antigen exposure in childhood may play a unique role in disease development [9].

Despite growing insights into these mechanisms, effective and standardized treatments for juvenile scleroderma remain limited. Pediatric-specific clinical trials are scarce, and most therapeutic decisions are still based on adult data. This narrative review aims to summarize the current knowledge on emerging systemic therapies for jLS and jSS, highlighting new biologics, small molecule agents, and advanced therapeutic strategies that are under investigation, with the goal of supporting more targeted and effective treatment approaches, tailored to the pediatric population.

## 2. Methods

This review aimed to synthesize emerging systemic therapies for juvenile scleroderma by analyzing studies published between 2001 and 2024, a period chosen to reflect two decades of significant therapeutic advances and increasing pediatric focus. Figure 1 shows the selection process of publications. A total of 52 studies were included, comprising randomized controlled trials, systematic reviews, and case reports; the latter were incorporated to capture rare pediatric presentations and preliminary evidence where higher-level data are lacking. Although case reports offer a lower evidence strength, they provide valuable clinical insights, especially for refractory or severe disease forms. Only English-language articles were considered, due to consistency in evaluating the data quality, although this may have introduced some selection bias. The following combinations of keywords were used: [(juvenile scleroderma) OR (juvenile systemic sclerosis)] AND [(biological therapy) OR (targeted therapy)].

Additionally, a manual search of the bibliography in the identified articles was also performed for the potential inclusion of further papers that would help keep the results of our search up to date. To complement our research, we also searched individually for each specific drug related to this clinical condition to ensure the inclusion of all papers of interest.

## 3. Results

### 3.1. Juvenile Localized Scleroderma (jLS)

No definite cure is available for either adult localized scleroderma or jLS. However, the disease severity may influence the therapeutic choices in children: in particular, a “low disease severity” is defined as a circumscribed superficial morphea that is not associated with subcutaneous atrophy, as well as extra-cutaneous involvement or scalp hair loss; a “moderate-to-high disease severity” includes all other subtypes, including deeper tissue lesions [10]. The management of localized scleroderma primarily focuses on reducing inflammation to prevent progression and minimize skin and tissue damage. However, no treatments are currently approved specifically for this condition, making therapeutic decisions largely experience-based. Depending on the disease severity, the treatment options vary: milder cases are generally addressed with topical therapies, including corticosteroids, tacrolimus, imiquimod, vitamin D derivatives, or phototherapy [10]. Conversely, moderate-to-severe forms can be treated with immunosuppressive medications, choosing a methotrexate-based regimen as the first-line therapy (with or without intravenous or oral corticosteroids) [10]. Mycophenolate mofetil may be used in relapsing cases or in methotrexate-refractory ones [11]. Further agents, including cyclosporine, hydroxychloroquine, azathioprine, retinoids, intravenous immunoglobulin, rituximab, and infliximab, have all shown effectiveness in the most severe cases of pediatric morphea [12,13,14,15,16,17]. However, their routine use requires caution and long-term monitoring. Emerging management strategies in refractory cases of jLS include tocilizumab, sarilumab, abatacept, imatinib, pamrevlumab, Janus kinase (JAK) inhibitors, and autologous stem cell transplantation (ASCT). The text below and Table 2 report the main treatments and their mechanism of action, the current evidence, their pediatric relevance, their safety profile, and the main available studies.

#### 3.1.1. Tocilizumab and Sarilumab

Tocilizumab and sarilumab are fully humanized monoclonal antibodies that block IL-6 binding to soluble IL-6 receptor, thereby inhibiting IL-6 signaling, a crucial pathway in the pathogenesis of scleroderma, which regulates fibroblast activity, stimulates collagen production, and inhibits the synthesis of collagenases. Recently, there has been a growing interest in the use of tocilizumab, as considerably high serum levels of IL-6 have been found in patients with localized scleroderma [18]. In 2017, a case series reported 11 children with jLS, 3 of whom were treated with intravenous tocilizumab at the dose of 8 mg/kg every 3-4 weeks, according to the available experience in the treatment of juvenile idiopathic arthritis [19]. The success after tocilizumab was reflected by improved measurements of activity index and skin damage index, and no complications were reported [20]. Furthermore, based on a randomized placebo-controlled trial of subcutaneous tocilizumab in adult systemic sclerosis which demonstrated a skin thickness reduction compared with placebo [21], Lythgoe et al. hypothesized that tocilizumab might also be beneficial for children with severe refractory jLS: 5 patients with longstanding active disease, nonresponsive to multiple treatments, were treated with tocilizumab at the dose of 8 mg/kg (in children weighing >30 kg) and 12 mg/kg (for those weighing <30 kg) at weeks 0, 2, and 4, and then every 4 weeks; after 12-25 months of treatment, all patients remained on tocilizumab, showing some improvement in disease activity, particularly in the physician’s global assessment of activity. However, there were no improvements in disease damage indices and quality of life. Except for 1 patient who was re-started on methotrexate and mycophenolate mofetil, the others continued to receive tocilizumab as monotherapy. No serious adverse reactions were reported [22]. Further 3 case reports demonstrated tocilizumab effectiveness in reducing disease activity for children with resistant pansclerotic morphea, the most severe subtype of jLS, suggesting that this agent could be used earlier in the disease course, in any case before occurrence of extensive sclerosing damage [23,24,25]. Hence, tocilizumab appears to be a valuable option for treatment-resistant jLS based on the favourable results obtained in studies related to adult systemic sclerosis [21] as well as on overall good tolerability and efficacy shown in children [20], though further studies are needed to confirm its role.

Sarilumab is a similar monoclonal antibody, binding IL-6 receptor in vitro with 15-to-22 fold higher affinity than tocilizumab, which is currently approved for the treatment of moderate-to-severe rheumatoid arthritis in non-responsive adults [26]: a phase 2 open-label clinical trial (NCT03679845) was started to evaluate sarilumab in halting the progression of morphea, but it was withdrawn due to the difficulty in recruiting all-aged patients [27]. To clarify the impact of sarilumab on the progression of linear scleroderma requires further observational studies.

#### 3.1.2. Abatacept

Abatacept is a first-generation recombinant cytotoxic T-lymphocyte-associated antigen 4 (CTLA-4) immunoglobulin that acts as a T cell costimulation blocker, having the power to inhibit T lymphocyte activation, which was proved to be a significantly effective treatment in adults with localized scleroderma [28]. A retrospective case series of 8 patients with jLS, treated with abatacept (intravenously administered at 10 mg/kg at days 0, 14 and 28, then once monthly) combined with methylprednisolone and mycophenolate mofetil or methotrexate was reported in 2020. All patients had received intravenous methylprednisolone at the start of treatment, making it unclear whether abatacept alone could have been effective; however, these patients had previously failed different treatments, suggesting that abatacept may have contributed to their improvement [29]. Future controlled trials are needed to confirm abatacept efficacy and to determine whether it acts synergistically with other drugs.

#### 3.1.3. Imatinib

Many molecular and biological evidence highlight a key-role of the platelet-derived growth factor (PDGF) receptor, a tyrosine kinase-associated receptor, in the development of systemic sclerosis: in particular, imatinib is an anti-fibrotic drug belonging to the tyrosine kinase inhibitor family, that interferes with both transforming growth factor beta (TGF-β) and PDGF signaling pathways by blocking the activity of c-Abl, c-Kit, and PDGF receptors, respectively [30]. This drug has demonstrated beneficial results in adult patients with morphea [31,32,33], while a phase 2 randomized clinical trial (NCT00479934) is still ongoing. Based on in vitro studies [30] and clinical response observed in case reports, it is expected that imatinib might be also effective in the pediatric population of patients. In 2013, Inamo and Ochiai reported one 3-year-old child with progressive jLS treated with a combination of imatinib, systemic corticosteroids, and methotrexate: this therapy halted the progressive skin thickening and also patient’s hand and finger joint deformities, concluding that imatinib may be used in children with jLS together with corticosteroids and immunosuppressant drugs [34]. 

#### 3.1.4. Janus Kinase Inhibitors (JAK)

JAK inhibitors, though not yet approved for pediatric patients, are thought to be beneficial for treating different interferon-mediated inflammatory disorders, and are increasingly being reported as therapeutic options for treatment-resistant morphea based on preclinical trials on skin fibrosis [35]. Two cases of jLS were successfully treated with tofacitinib: they were a 6-year-old girl and a 13-year-old boy who showed significant improvement after 6 months of tofacitinib (after failure of methotrexate due to side effects or inadequate compliance). These patients on tofacitinib were shown to have skin lesions softened, sclerosis resolved, and histological improvement was also confirmed; no significant adverse effects were reported [36]. In contrast, another pediatric patient with a pansclerotic morphea showed no response to ruxolitinib, despite having previously failed several treatments [37]. The improvement of fibrosis could be explained by a comparative study describing that tofacitinib inhibited the fibrotic pathway by blocking TGF-β-mediated activation of signal transducer and activator of transcription (i.e. STAT) protein [38]. While spontaneous regression of morphea cannot be excluded, these studies highlight the potential of tofacitinib as a therapeutic tool in treatment-refractory pediatric morphea, though further studies are required to assess its long-term efficacy and also its safety.

#### 3.1.5. Autologous Stem Cell Transplantation (ASCT)

ASCT is a potential therapeutic option for severe refractory linear scleroderma, which has been performed in 2 children with disabling morphea unresponsive to systemic therapies. Moll et al. used a CD3/CD19-depleted graft after immunoablative conditioning with fludarabine, cyclophosphamide and anti-thymocyte globulin, which led to improvement of skin lesions and limb activity. However, the disease relapsed, leaving unclear whether continuing immunosuppressive therapy post-ASCT would provide any long-term benefit [39]. 

### 3.2. Juvenile Systemic Sclerosis (jSS)

Although the underlying mechanisms driving different forms of scleroderma are not yet fully understood, abnormal collagen accumulation remains central to the development of fibrosis in both the skin and internal organs. Currently, no treatment for systemic sclerosis has demonstrated clear, consistent disease-modifying effects. In pediatric cases, therapeutic strategies are largely based on adult studies due to the limited availability of child-specific clinical data, emphasizing the need for global collaborative efforts to establish evidence-based pediatric interventions. However, current therapeutic interventions should be initiated early in the disease course to maximize any beneficial effect. The most recent guidelines for pediatric rheumatology recommend methotrexate as a first-line treatment strategy in jSS, especially for treating skin, osteoarticular, gastrointestinal, and vascular manifestations, whereas mycophenolate mofetil is more often suggested in the most difficult-to-treat and resistant cases [4]. Conversely, cyclophosphamide is recommended for SS-related interstitial lung and cardiac disease [4]. Novel interventions such as biotechnological therapies, including rituximab, tocilizumab, JAK inhibitors, abatacept, imatinib, nintedanib, and ASCT, are under investigation. The text below and Table 3 report the main treatments and their mechanism of action, the current evidence, their pediatric relevance, their safety profile, and the main available studies.

#### 3.2.1. Rituximab

Rituximab is a chimeric monoclonal antibody that targets the CD20 antigen on B-cells, leading to B-cell depletion and suppression of autoantibody production. In a small observational study, Zulian et al. treated four pediatric patients with jSS using rituximab combined with mycophenolate mofetil over a one-year period [40]. All patients demonstrated a reduction in the frequency and duration of Raynaud’s phenomenon and a decrease in skin involvement. Additionally, two patients showed improvement in cardiac function, while the remaining two experienced better respiratory performance [40].

This therapy has been used off-label in children, with the evidence currently being limited to case series. The level of evidence remains low due to the small sample size and absence of control groups.

No serious adverse events were reported in this study. However, pediatric safety data remain scarce. A systematic review by Kaegi et al. confirmed beneficial outcomes in systemic sclerosis for lung and skin involvement but emphasized the need for more robust safety data, particularly in children [41].

#### 3.2.2. Tocilizumab

Tocilizumab is a humanized monoclonal antibody against the IL-6 receptor. It blocks IL-6-mediated signaling, which plays a pivotal role in inflammation, immune activation, and fibrosis. In a retrospective pilot study by Adrovic et al., nine pediatric patients with jSS were treated with tocilizumab in addition to standard therapies [42]. The patients showed significant improvements in the modified Rodnan skin score, lung diffusion capacity (DLCO), thoracic HRCT findings, and composite jSS disease scores. The treatment was well tolerated [42]. A separate case report described a 13-year-old girl with extensive multi-organ involvement who experienced clinical remission with tocilizumab monotherapy, without adverse events [43].

Although the data are promising, the current evidence is limited to observational studies and case reports, representing a moderate-to-low level of evidence.

In both the pilot study and case report, tocilizumab demonstrated a favorable safety profile in children. However, long-term safety data are needed to support broader clinical use.

#### 3.2.3. Janus Kinase (JAK) Inhibitors

JAK inhibitors disrupt cytokine signaling by inhibiting Janus kinase enzymes, which are essential mediators of inflammation and fibrosis in autoimmune diseases.

Pin et al. reported the off-label use of tofacitinib in seven pediatric patients with severe inflammatory conditions, including a 13-year-old girl with jSS [44]. This patient, who was refractory to corticosteroids, mycophenolate mofetil, and rituximab, showed marked clinical improvement within one month, including enhanced joint mobility and normalization of inflammatory markers. The clinical benefits were maintained for three months [44].

The use of JAK inhibitors in jSS is experimental and currently only supported by anecdotal evidence. The level of evidence is very low, and controlled pediatric studies are lacking.

The reported case did not reveal adverse events, but comprehensive safety assessments in pediatric populations are still required due to potential risks such as infections and growth effects.

#### 3.2.4. Abatacept

Abatacept is a recombinant fusion protein that inhibits T-cell activation by binding CD80/86 and preventing costimulatory signaling via CD28, a key mechanism in autoimmune pathogenesis [45].

Preclinical studies demonstrated that abatacept reduced fibrosis in mouse models of systemic sclerosis by modulating T-cell activity [46,47]. In a 12-month randomized, double-blind, placebo-controlled trial in adults with early diffuse cutaneous systemic sclerosis, Khanna et al. found that while abatacept did not significantly improve the Rodnan skin score, secondary endpoints such as gene expression profiles suggested potential therapeutic benefits [48].

No pediatric trials have evaluated abatacept in jSS to date. All supporting data are based on adult studies and animal models, representing a low level of evidence for pediatric use.

The adult study reported good tolerability, but the pediatric safety profile in jSS remains undetermined.

#### 3.2.5. Imatinib and Nintedanib

Imatinib is a tyrosine kinase inhibitor that blocks the PDGF and TGF-β signaling pathways, which are both central to fibrogenesis. Nintedanib also inhibits multiple tyrosine kinases, including VEGFR, FGFR, and PDGFR, and is used in fibrosing lung conditions. Imatinib has shown efficacy in adult patients with systemic sclerosis, improving Rodnan skin scores according to a meta-analysis [49]. However, its impact on lung disease remains inconclusive. Nintedanib has demonstrated the ability to stabilize lung function in adults with systemic sclerosis-related interstitial lung disease, as reported by Boutel et al. [50].

Neither agent has been studied in children with jSS. Therefore, the evidence remains indirect and extrapolated from adult data, classifying it as very low-level evidence in pediatric populations.

While its tolerability in adults has been acceptable, potential adverse effects such as gastrointestinal intolerance and liver enzyme elevations raise safety concerns for pediatric use.

#### 3.2.6. Pamrevlumab

Pamrevlumab is a monoclonal antibody targeting connective tissue growth factor (CTGF), a key mediator in fibrogenic signaling and tissue remodeling [51].

In a murine model of skin fibrosis induced by angiotensin II, Makino et al. demonstrated that pamrevlumab significantly reduced skin fibrosis, supporting its potential antifibrotic role in systemic sclerosis [52]. No clinical trials, case reports, or pediatric studies have been conducted. The evidence remains entirely preclinical.

While promising in animal models, the safety of pamrevlumab in both adult and pediatric systemic sclerosis is yet to be determined.

#### 3.2.7. Autologous Stem Cell Transplantation (ASCT)

ASCT involves high-dose immunosuppressive therapy, followed by the reinfusion of autologous hematopoietic stem cells to reset the immune system and eliminate autoreactive lymphocytes.

Only one case report describes ASCT in a pediatric patient with severe refractory jSS. The adolescent girl received conditioning therapy followed by CD52-chimeric monoclonal antibody infusion, resulting in sustained clinical improvement [53].

The use of ASCT in children with jSS is extremely rare and supported by very low-level evidence. It may be considered only in life-threatening, treatment-refractory cases.

ASCT carries significant risks, including infections, infertility, and secondary malignancies. Careful patient selection is essential, especially in pediatric patients [4].

#### 3.2.8. Chimeric Antigen Receptor (CAR-T) Cell Therapy

CAR-T cell therapy uses genetically engineered T-cells to express a receptor targeting CD19+ B-cells, aiming to eliminate autoreactive B-cells and restore immune balance. Originally developed for hematologic malignancies [54], CAR-T cells have shown promise in autoimmune disease models. Kansal et al. demonstrated disease remission in murine models of SLE following CD19 CAR-T therapy [55,56,57,58]. In a recent case series, Müller et al. treated adult patients with SLE, idiopathic inflammatory myositis, and systemic sclerosis, achieving long-term drug-free remission. Patients with systemic sclerosis experienced improvements in skin and lung symptoms [59].

No pediatric data are currently available for CAR-T therapy in jSS. The evidence remains preclinical and early-phase adult clinical, classifying it as very low-level for pediatric systemic sclerosis.

While effective, CAR-T therapy is associated with risks such as cytokine release syndrome and neurotoxicity. Its safety in pediatric autoimmune diseases remains to be established through clinical trials.

## 4. Conclusions

The management of juvenile scleroderma continues to pose significant challenges due to the disease’s clinical heterogeneity, unpredictable course, and the lack of standardized, pediatric-specific therapeutic guidelines. Treatment decisions must be individualized, taking into account the specific subtype (jLS or jSS), disease severity, and extent of organ involvement. Early and aggressive intervention remains crucial to reduce the risk of irreversible complications such as disfigurement, functional disability, and progressive visceral fibrosis.

For jLS, current strategies range from topical therapies (such as corticosteroids and calcineurin inhibitors) for mild cases to systemic immunosuppressants like methotrexate and mycophenolate mofetil in moderate-to-severe disease. In refractory cases, biologics including tocilizumab, sarilumab, and tofacitinib, as well as small molecule inhibitors like imatinib, offer promising alternatives. However, the available data are primarily derived from small case series or anecdotal reports, with limited controlled evidence. ASCT is an emerging option for severe, treatment-resistant cases but carries significant risk and is currently reserved for carefully selected patients.

In jSS, methotrexate and mycophenolate mofetil remain the mainstays of therapy. Biologic agents such as rituximab, tocilizumab, and abatacept, along with experimental treatments like JAK inhibitors, nintedanib, pamrevlumab, and CAR-T cell therapy, are under investigation. Nonetheless, the current evidence base is weak, often extrapolated from adult studies, and lacks the statistical robustness required for pediatric-specific recommendations.

Figure 2 summarizes the treatment strategy for juvenile scleroderma.

A key limitation across all emerging therapies is the scarcity of high-quality evidence, particularly the absence of large-scale, randomized controlled trials in pediatric populations. Most existing studies suffer from small sample sizes, retrospective designs, and heterogeneous outcome measures. To address these gaps, future research must prioritize the following: (1) multicenter, prospective clinical trials focused on pediatric populations; (2) the standardization of outcome measures across studies for comparability; (3) long-term follow-up to assess the durability of responses and safety; (4) exploration of personalized treatment algorithms based on molecular or immunologic subtypes; (5) integration of innovative approaches like CAR-T cell therapy in early-phase pediatric trials. Such efforts are essential to transform current anecdotal treatment paradigms into evidence-based, individualized care strategies for children with juvenile scleroderma.

## Figures and Tables

**Figure 1 pharmaceuticals-18-00643-f001:**
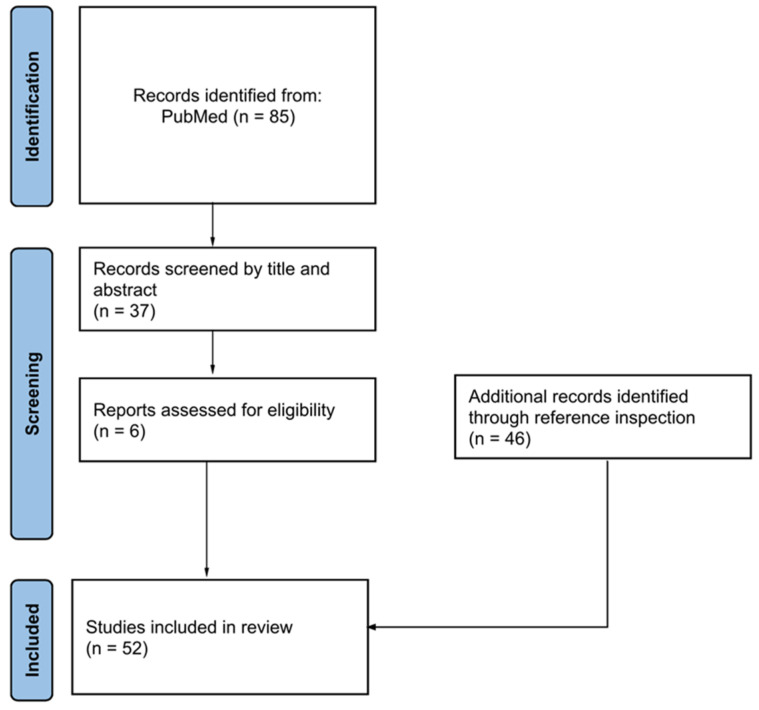
Description of the selection process of publications reporting on biologic treatments for juvenile scleroderma in our review.

**Figure 2 pharmaceuticals-18-00643-f002:**
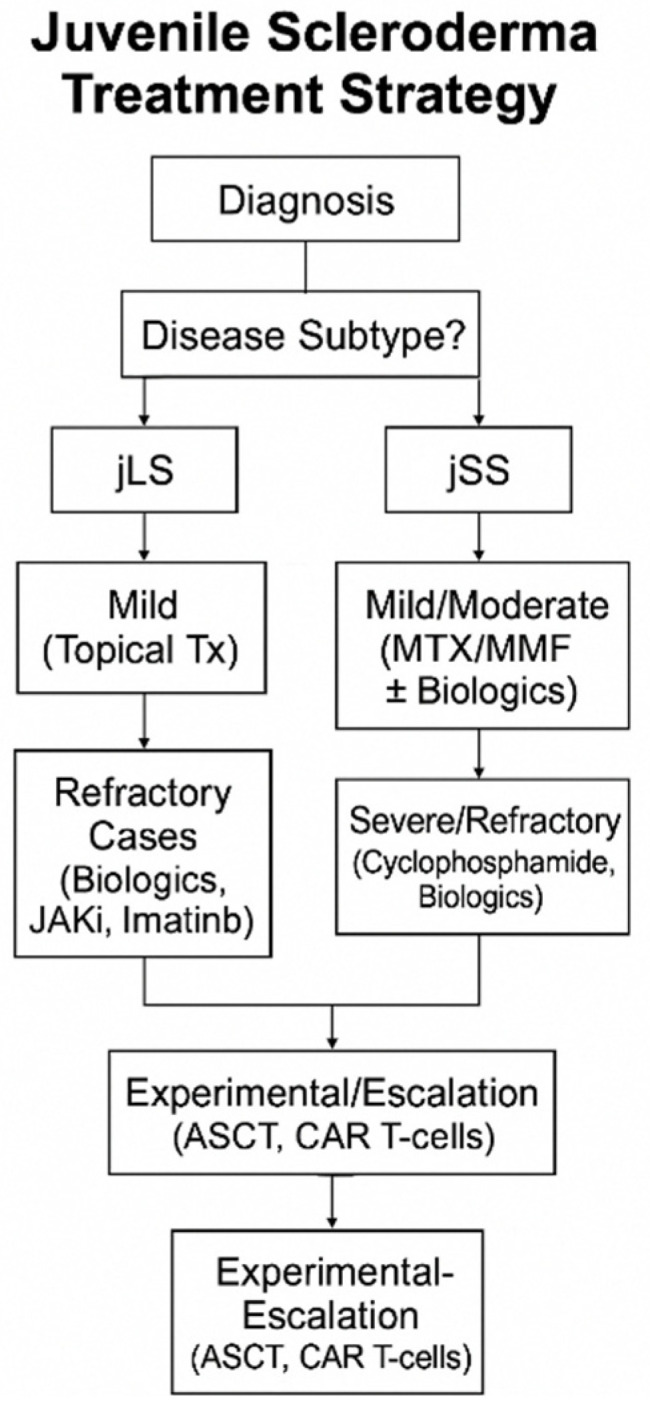
Flow diagram of the treatment strategy for juvenile scleroderma. ASCT: autologous stem cell transplantation; CAR-T cells: chimeric antigen receptor T-cells; jLS: juvenile localized scleroderma; jSS: juvenile systemic sclerosis; MMF: mycophenolate mofetil; MTX: methotrexate; JAKi: janus kinase inhibitors; Tx: treatment.

**Table 1 pharmaceuticals-18-00643-t001:** Classification criteria for juvenile systemic sclerosis in childhood, developed in 2007 to standardize clinical research, epidemiologic studies, and therapeutic trials.

Major Criterion (Required)	Proximal Sclerosis/Induration of the Skin	
**Minor criteria** (involving specific districts of the body)	Skin	Sclerodactyly
	Vascular	Raynaud’s phenomenon
		Nailfold videocapillaroscopy abnormalities
		Digital tip ulcers
	Gastrointestinal	Dysphagia
		Gastroesophageal reflux
	Renal	Renal crisis
		New-onset arterial hypertension
	Cardiac	Arrhythmias
		Heart failure
	Respiratory	Pulmonary fibrosis (seen on HCRT or radiography of chest)
		Abnormal diffusing capacity of the lungs for carbon monoxide
		Pulmonary hypertension
	Musculoskeletal	Tendon friction rubs
		Arthritis
		Myositis
	Neurologic	Neuropathy
		Carpal tunnel syndrome
	Serology	Anti-nuclear antibodies
		Selective autoantibodies for systemic sclerosis (anti-centromere, anti-topoisomerase I, anti-fibrillarin, anti-PM/Scl, anti-fibrillin, or anti-RNA polymerase I/III antibodies)

**Table 2 pharmaceuticals-18-00643-t002:** Overview of biologic treatments for children with juvenile localized scleroderma.

Drug	Number of Patients	Article (Year)
**Tocilizumab**		
8 mg/kg every 3–4 weeks SC	3	#19 (2017)
8 mg/kg (>30 kg) SC 12 mg/kg (<30 kg) SC weeks 0, 2, and 4, and then every 4 weeks	5	#22 (2018)
8 mg/kg every 4 weeks SC	2	#23 (2017)
300 mg IV every 4 weeks	1	#24 (2019)
**Abatacept**		
10 mg/kg SC at days 0, 14 and 28, then once monthly	8	#29 (2020)
**Imatinib**		
235 mg/m^2^/day per os	1	#34 (2013)
**JAK inhibitors**		
**Tofacitinib**: 2.5 mg twice daily for two months, then once daily for 4 months then every other day, for 2 months per os	1	#38 (2023)
**Tofacitinib**: 5 mg, twice daily for 2 month, then 5 mg once daily per os	1	#38 (2023)
**Ruxolitinib**: 10 mg twice daily per os	1	#39 (2019)

Abbreviations: SC, subcutaneously; IV, intravenously.

**Table 3 pharmaceuticals-18-00643-t003:** Overview of biologic treatments in children with juvenile systemic sclerosis.

Drug	Number of Patients	Article (Year)
**Rituximab**		
375 mg/m^2^ IV on day 0 and 14, at 3-month intervals	4	#40 (2020)
**Tocilizumab**		
8 mg/kg (children ≥ 30 kg) and 12 mg/kg (children < 30 kg) SC once every 4 weeks	9	#42 (2021)
162 mg SC once every 2 weeks	1	#43 (2022)
**JAK inhibitors**		
**Tofacitinib**: 5 mg twice per day	1	#44 (2020)
**Abatacept**		
125 mg SC	88	#48 (2020)
**Imatinib and nintedanib**		
**Imatinib**: 200–400 mg/day orally used in adults in several early-phase studies; 235 mg/m^2^/day orally, adjusted to body surface area, in a pediatric case report	Unknown	#49 (2020)
**Nintedanib**: Approved adult dose: 150 mg orally twice daily (total daily dose: 300 mg); dose adjustments for children have not been formally defined due to lack of pediatric trials	Unknown	#50 (2023)
**Pamrevlumab**		
Investigational dose in adults in clinical trials: 30 mg/kg intravenously every 3 weeks; no clinical data available for pediatric populations	Mice models	#52 (2017)

Abbreviations: SC, subcutaneously; IV, intravenously.

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
