# Peer review of "New Challenging Systemic Therapies for Juvenile Scleroderma: A Comprehensive Review"

_pharmaceuticals, 2025, doi:10.3390/ph18050643_

Round 1
Reviewer 1 Report
Comments and Suggestions for Authors
This well-conceived and timely review provides a comprehensive overview of emerging systemic therapies for juvenile scleroderma (both jLS and jSS). The breadth of therapies covered, ranging from conventional immunomodulators to advanced treatments such as JAK inhibitors, CAR-T cells, and autologous stem cell transplantation, reflects the current evolution of the therapeutic landscape in pediatric rheumatology.
I would recommend some minor revisions to enhance clarity, consistency, and readability:
- Language and Style
- Please revise for clarity and fluency. Several sections contain long or complex sentences that would benefit from simplification.
- Structure and Consistency
- Some treatment sections (e.g., tocilizumab, abatacept, and JAK inhibitors) vary in format and detail. Consider standardizing the structure across all drug discussions: mechanism of action, current evidence, pediatric relevance, and safety profile.
- Uniform inclusion of dosing information (where available) would enhance utility for clinicians.
- Critical Appraisal
- In sections where therapies are discussed based on limited pediatric data or case reports, please indicate the level of evidence and any limitations more clearly.
- Consider adding a graphical summary or treatment flow diagram to visually synthesize treatment options or proposed escalation strategies.
- Conclusion
- The conclusion is appropriately balanced but could be strengthened with a brief forward-looking statement—e.g., highlighting the need for multicenter trials or more pediatric-focused studies.
The manuscript is generally well-written and communicates the scientific content effectively. However, there are several instances of grammatical errors, awkward phrasing, and overly complex sentence structures that affect readability. A thorough round of language editing is recommended to improve clarity, sentence flow, and consistency.
Consider simplifying long sentences, ensuring subject-verb agreement, and using consistent tenses throughout the manuscript.
Author Response
This well-conceived and timely review provides a comprehensive overview of emerging systemic therapies for juvenile scleroderma (both jLS and jSS). The breadth of therapies covered, ranging from conventional immunomodulators to advanced treatments such as JAK inhibitors, CAR-T cells, and autologous stem cell transplantation, reflects the current evolution of the therapeutic landscape in pediatric rheumatology.
I would recommend some minor revisions to enhance clarity, consistency, and readability:
Re: Thank you very much for your positive evaluation. We revised our manuscript according to the suggestions.
Language and Style
Please revise for clarity and fluency. Several sections contain long or complex sentences that would benefit from simplification.
Re: Simplified and improved as suggested.
Structure and Consistency
Some treatment sections (e.g., tocilizumab, abatacept, and JAK inhibitors) vary in format and detail. Consider standardizing the structure across all drug discussions: mechanism of action, current evidence, pediatric relevance, and safety profile.
Re: Revised accordingly (pp. 4-7 and 8-10).
Uniform inclusion of dosing information (where available) would enhance utility for clinicians.
Re: Revised accordingly (pp. 4-11).
Critical Appraisal
In sections where therapies are discussed based on limited pediatric data or case reports, please indicate the level of evidence and any limitations more clearly.
Re: Done (pp. 4-7 and 8-10).
Consider adding a graphical summary or treatment flow diagram to visually synthesize treatment options or proposed escalation strategies.
Re: Done (p. 11)
Conclusion
The conclusion is appropriately balanced but could be strengthened with a brief forward-looking statement—e.g., highlighting the need for multicenter trials or more pediatric-focused studies.
Re: Conclusions have been rewritten according to your comments.
Comments on the Quality of English Language
The manuscript is generally well-written and communicates the scientific content effectively. However, there are several instances of grammatical errors, awkward phrasing, and overly complex sentence structures that affect readability. A thorough round of language editing is recommended to improve clarity, sentence flow, and consistency.
Consider simplifying long sentences, ensuring subject-verb agreement, and using consistent tenses throughout the manuscript.
Re: Thank you for the suggestions. The text has been improved by an English mother tongue with appropriate knowledge of the subject’s matter.
Reviewer 2 Report
Comments and Suggestions for Authors
This review provides a comprehensive and up-to-date overview of systemic therapies for juvenile scleroderma (JS), encompassing both juvenile localized scleroderma (JLS) and juvenile systemic sclerosis (jSS). The manuscript is well-structured, clearly written, and addresses a significant gap in pediatric rheumatology. However, several areas could be improved to enhance clarity, depth, and clinical applicability.
- Please clarify the timeframe of the literature search (e.g., "2001–2024" is mentioned, but the rationale for this range is not explained).
- Please specify the number of studies included in the review (e.g., "52 studies" as per Figure 1) to provide context for the evidence base.
- Expand on the “pathophysiological differences” between JLS and jSS, as this is critical for understanding therapeutic targets.
- You need to briefly mention the burden of disease (e.g., quality of life, long-term morbidity) to underscore the urgency of novel treatments.
- Clarify why “case reports” were included alongside randomized trials and reviews, given their lower evidence level. Consider stratifying results by study type.
- Justify the exclusion of non-English studies, as this may introduce bias.
- I suggest adding columns for outcomes (e.g., "improved skin score," "reduced disease activity"), and adverse effects in Table 2 to facilitate comparison.
- Add a subheading, highlighting the limitation of the current evidence because of small sample sizes and lack of controlled trials for most biologics.
- It is better to include the dosing regimens for drugs like nintedanib and pamrevlumab in Table 3, even if the data are from adults.
- In Fig. 1, label the boxes more clearly (e.g., "Records identified," "Excluded").
- I suggest considering a summary table comparing the efficacy/safety of all therapies for JLS vs. jSS.
- For a review paper, min. number of citations should be more than 100. Please check the journal author guidelines.
- Please define abbreviations at first use (e.g., "ASCT" in Abstract). Meanwhile, there are minor typos (e.g., "mycophenolate more than 5% of the serum" → likely "mycophenolate mofetil"). Finally, use uniform terminology for drug administration routes (e.g., "SC" vs. "subcutaneously").
Please define abbreviations at first use (e.g., "ASCT" in Abstract). Meanwhile, there are minor typos (e.g., "mycophenolate more than 5% of the serum" → likely "mycophenolate mofetil"). Finally, use uniform terminology for drug administration routes (e.g., "SC" vs. "subcutaneously").
Author Response
This review provides a comprehensive and up-to-date overview of systemic therapies for juvenile scleroderma (JS), encompassing both juvenile localized scleroderma (JLS) and juvenile systemic sclerosis (jSS). The manuscript is well-structured, clearly written, and addresses a significant gap in pediatric rheumatology. However, several areas could be improved to enhance clarity, depth, and clinical applicability.
Re: Thank you for your positive evaluation. We revised the manuscript according to your comments.
- Please clarify the timeframe of the literature search (e.g., "2001–2024" is mentioned, but the rationale for this range is not explained).
Re: Clarified (p. 3).
- Please specify the number of studies included in the review (e.g., "52 studies" as per Figure 1) to provide context for the evidence base.
Re: Specified (p. 3).
- Expand on the “pathophysiological differences” between JLS and jSS, as this is critical for understanding therapeutic targets.
Re: Done (p. 3).
- You need to briefly mention the burden of disease (e.g., quality of life, long-term morbidity) to underscore the urgency of novel treatments.
Re: Done (p. 2).
- Clarify why “case reports” were included alongside randomized trials and reviews, given their lower evidence level. Consider stratifying results by study type.
Re: Clarified (p. 3).
- Justify the exclusion of non-English studies, as this may introduce bias.
Re: Justified (p. 3).
- I suggest adding columns for outcomes (e.g., "improved skin score," "reduced disease activity"), and adverse effects in Table 2 to facilitate comparison.
Re: There aren’t enough data to add outcomes in the Table. However,
- Add a subheading, highlighting the limitation of the current evidence because of small sample sizes and lack of controlled trials for most biologics.
Re: We revised the Conclusions including this comment (pp. 11-12).
- It is better to include the dosing regimens for drugs like nintedanib and pamrevlumab in Table 3, even if the data are from adults.
Re: Added as requested (pp. 10-11).
- In Fig. 1, label the boxes more clearly (e.g., "Records identified," "Excluded").
Re: We revised the text improving the clarity of Figure 1.
- I suggest considering a summary table comparing the efficacy/safety of all therapies for JLS vs. jSS.
Re: According to other reviewer’s suggestions, we added a flow-diagram on treatment strategies (p. 12).
- For a review paper, min. number of citations should be more than 100. Please check the journal author guidelines.
Re: This is an innovative manuscript. We included all the references identified through our search strategy.
- Please define abbreviations at first use (e.g., "ASCT" in Abstract). Meanwhile, there are minor typos (e.g., "mycophenolate more than 5% of the serum" → likely "mycophenolate mofetil"). Finally, use uniform terminology for drug administration routes (e.g., "SC" vs. "subcutaneously").
Re: Revised accordingly.
Comments on the Quality of English Language
Please define abbreviations at first use (e.g., "ASCT" in Abstract). Meanwhile, there are minor typos (e.g., "mycophenolate more than 5% of the serum" → likely "mycophenolate mofetil"). Finally, use uniform terminology for drug administration routes (e.g., "SC" vs. "subcutaneously").
Re: Revised as suggested.